# Adaptation-Agnostic Meta-Training

**Jiaxin Chen** *                                    JIAX.CHEN@CONNECT.POLYU.HK
**Li-Ming Zhan**\*                                   LMZHAN.ZHAN@CONNECT.POLYU.HK
**Xiao-Ming Wu** †                                   XIAO-MING.WU@POLYU.EDU.HK
**Fu-Lai Chung** †                                   CSKCHUNG@POLYU.EDU.HK
*The Hong Kong Polytechnic University*

## Abstract

Many meta-learning algorithms can be formulated into an interleaved process, in the sense that task-specific predictors are learned during inner-task adaptation and meta-parameters are updated during meta-update. The normal meta-training strategy needs to differentiate through the inner-task adaptation procedure to optimize the meta-parameters. This leads to a constraint that the inner-task algorithms should be solved analytically. Under this constraint, only simple algorithms with analytical solutions can be applied as the inner-task algorithms, limiting the model expressiveness. To lift the limitation, we propose an adaptation-agnostic meta-training strategy. Following our proposed strategy, we can apply stronger algorithms (e.g., an ensemble of different types of algorithms) as the inner-task algorithm to achieve superior performance comparing with popular baselines. The source code is available at https://github.com/jiaxinchen666/AdaptationAgnosticMetaLearning.

## 1. Introduction

Meta-learning is a promising solution to endow machines with skills to fast adapt to new environments with few experiences. From a unified view, the commonly used meta-training procedure in existing meta-algorithms is an interleaved process which includes *inner-task adaptation* and *meta-update*. During inner-task adaptation, the inner-task algorithm $\mathcal{A}$ runs through the support set $\mathcal{D}_{T_i}^{tr}$ and outputs a predictor $g_{\phi_{T_i}}$ parameterized by *task-specific parameters* $\phi_{T_i}$. During meta-update, the loss of the task-specific predictor over the query set $\mathcal{D}_{T_i}^{ts}$ is minimized to update the *meta-parameters* $\theta$ that are shared by all tasks. The update rule of meta-parameters is formulated as follows.

$$\theta = \theta - \nabla_\theta \mathcal{L}(\mathcal{D}_{T_i}^{ts}; \phi_{T_i}), \text{ where } g_{\phi_{T_i}} = \mathcal{A}(\mathcal{D}_{T_i}^{tr}; \theta). \tag{1}$$

In this formulation, the optimization of meta-parameters should differentiate through the inner-task adaptation. To obtain an explicit and differentiable meta-objective function and its gradient w.r.t. $\theta$ in Eq. (1), we should compute $\nabla_\theta \phi_{T_i}$. Hence, the inner-task algorithm $\mathcal{A}$ should be solved analytically, i.e., $\phi_{T_i}$ should be an analytical expression of $\theta$,

$$\phi_{T_i} = s(\mathcal{D}_{T_i}^{tr}, \theta). \tag{2}$$

To satisfy this requirement, only simple algorithms with closed-form solvers can be applied as the inner-task algorithm, such as nearest neighbor classification (Snell et al., 2017), ridge

---

∗. Equal contribution
†. Corresponding authors

regression (Bertinetto et al., 2019), SVM (Lee et al., 2019) or gradient descent with a learned initialization (Finn et al., 2017), which significantly limits its expressive power.

To enrich the choices of inner-task algorithms and improve the expressive power of models, we propose an **A**daptation-**A**gnostic **M**eta-training strategy (A2M) to remove the analytical dependency between the task-specific parameters and the meta-parameters. For inner-task adaptation, we *fix* the meta-parameters and use the support set to optimize the task-specific parameters. For meta-update, we *fix* the task-specific parameters and optimize the meta-parameters using the query set. The meta-parameters are updated by minimizing the predictor's loss over the embedded query set. Without differentiating the inner-task optimization process, the meta-training strategy is agnostic to the inner-task algorithm which only needs the solution of $\phi_{T_i}$.

The generality and flexibility of the proposed meta-training strategy makes it easy to combine different types of algorithms as an ensemble inner-task algorithm to exploit their advantages and alleviate their drawbacks. We introduce an instantiation of A2M and conduct extensive experiments on standard or cross-domain few-shot classification tasks over *mini*Imagenet and CUB to evaluate its effectiveness. Experiments show A2M achieves superior performance with low computational cost in comparison with the popular baselines.

## 2. Related Work

**Meta-learning algorithms** can be broadly categorized based on the type of the inner-task algorithm, namely, metric-based, gradient-based, model-based and meta-algorithms with closed-form solutions. *Metric-based* meta-algorithms learn a mapping from the data space to an embedding space, where the inner-task algorithm is a comparison algorithm based on a similarity metric (Koch et al., 2015; Vinyals et al., 2016; Snell et al., 2017; Garcia and Bruna, 2018). Similarly, over the embedding space, *meta-algorithms with closed-form solutions* apply simple inner-task algorithms with a closed-form solution such as ridge regression (Bertinetto et al., 2019) or SVM (Lee et al., 2019). *Gradient-based meta-learning* (Finn et al., 2017) uses a gradient descent optimizer with a learned initialization of a deep neural network as the inner-task algorithm.

**Decoupled training strategies** have been explored in meta-learning or multi-task learning literature (Franceschi et al., 2018; Frans et al., 2017; Yang and Hospedales, 2016; Zintgraf et al., 2019; Rajeswaran et al., 2019). Franceschi et al. (2018) proposed a bilevel programming for hyperparameter optimization and meta-learning, which is essentially similar to our proposed training strategy. Similar training strategies are adopted in Frans et al. (2017) and Yang and Hospedales (2016) for reinforcement learning and multi-task learning respectively, which divided a network into shared layers and task-specific layers and updated them iteratively. However, the motivation and use of the training strategy in these works are totally different from ours. We motivate the decoupled training procedure from an adaptation-agnostic perspective for meta-learning, which leads to a flexible inner-task algorithm with high expressiveness. Rajeswaran et al. (2019) decouples the meta-training procedure by drawing implict gradient which can be seen as an approximation of MAML.

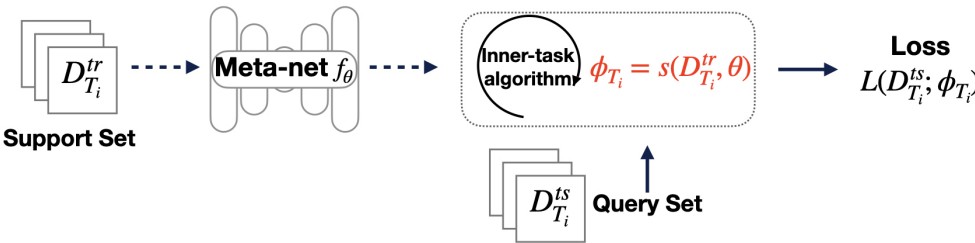

Figure 1: A common meta-training procedure of existing meta-algorithms. Note that $s(\cdot, \cdot)$ is an analytical expression.

## 3. A Unified View

### 3.1 Problem Formulation

The goal of a meta-algorithm $\mathbf{A}$ is to learn an *inner-task algorithm* $\mathcal{A}$ which can fast adapt to new tasks drawn from a task distribution $p(\tau)$. During meta-training, given a set of training tasks $\{T_i \sim p(\tau)\}_{i=1}^n$, the meta-algorithm observes the meta-samples $\mathbf{D} = \{\mathcal{D}_{T_i} = (\mathcal{D}_{T_i}^{tr}, \mathcal{D}_{T_i}^{ts})\}_{i=1}^n$, where $\mathcal{D}_{T_i}^{tr}$ is the training (*support*) set of task $T_i$ and $\mathcal{D}_{T_i}^{ts}$ is the test (*query*) set of $T_i$. Denote by $z = (x, y) \in \mathcal{Z} = (\mathcal{X}, \mathcal{Y})$ a data sample, where $\mathcal{X}$ is the feature space and $\mathcal{Y}$ is the label space. A support set and a query set per task are of size $m$ and $q$ respectively. After trained on these tasks, the meta-algorithm outputs a inner-task algorithm $\mathcal{A} = \mathbf{A}(\mathbf{D})$.

### 3.2 A Unified View of Existing Meta-Learning Methods

#### 3.2.1 META-TRAINING PROCEDURE

We characterize that the existing meta-algorithms leverage a meta-training procedure as the gradient of the meta-parameters $\theta$ is computed through the inner-task adaptation. As the meta-algorithm updates the meta-parameters $\theta$ by minimizing the loss of the task-specific predictor $g_{\phi_{T_i}}$ over the query set of each task. The update rule of $\theta$ is

$$\theta = \theta - \nabla_\theta \mathcal{L}(\{\mathcal{D}_{T_i}^{ts}\}_{i=1}^n; \theta, \{\phi_{T_i}\}_{i=1}^n) = \theta - \nabla_{\phi_{T_i}} \sum_{i=1}^n \sum_{z_j \in \mathcal{D}_{T_i}^{ts}} l\left(g_{\phi_{T_i}}(x_j), y_j\right) \times \nabla_\theta g_{\phi_{T_i}}, \quad (3)$$

where $g_{\phi_{T_i}} = \mathcal{A}(\mathcal{D}_{T_i}^{tr}; \theta)$ parameterized by task-specific parameters $\phi_{T_i}$. It can be discovered that following the update rule (3), the gradients of the meta-parameters are back-propagated though the task-specific parameters $\phi_{T_i}$. To make the back-propagation feasible, most existing meta-algorithms follow a common meta-training procedure as shown in Fig. 1. First, the task-specific parameters $\phi_{T_i}$ are directly computed w.r.t. $\theta$, i.e.,

$$g_{\phi_{T_i}} = \mathcal{A}(\mathcal{D}_{T_i}^{tr}; \theta), \text{where } \phi_{T_i} = s(\mathcal{D}_{T_i}^{tr}, \theta), \quad (4)$$

and $s(\cdot, \cdot)$ denotes an analytical expression. Then, $\phi_{T_i}$ is plugged back to the meta objective function (3) and $\theta$ is optimized by the gradient propagated from $\phi_{T_i}$. For example, the task-specific parameters of a typical **gradient-based meta-algotithm**, MAML (Finn et al.,

2017) $\phi_{T_i}$ is

$$\phi_{T_i} = \theta - l_{\phi_{T_i}} \nabla_\theta \mathcal{L}(\mathcal{D}_{T_i}^{tr}; \theta) = \theta - \nabla_\theta \big( \sum_{(x_j, y_j) \in \mathcal{D}_{T_i}^{tr}} l\big(f_\theta(x_j), y_j\big)\big). \tag{5}$$

Then, the gradients of $\theta$ include a second-order gradient of $\theta$, because $\nabla_\theta \phi_{T_i} = I - \nabla_\theta^2 \big( \sum_{(x_j, y_j) \in \mathcal{D}_{T_i}^{tr}} l\big(f_\theta(x_j), y_j\big)\big)$. It turns out that the choice of inner-task algorithm with different analytical expressions characterizes the key difference among existing meta-algorithms. Apart from gradient-based meta-algorithms such as MAML (Finn et al., 2017), the other popular meta-algorithms can be unified in this perspective.

**Metric-based meta-algorithms.** The inner-task algorithm of a metric-based meta-algorithm is a nearest neighbor algorithm with a distance function in the metric space, e.g., $d(x, x') = \|f_\theta(x) - f_\theta(x')\|_2^2$. For matching networks (Vinyals et al., 2016), the nearest neighbor algorithm is non-parametric, so there is no explicit training in inner-task adaptation. For prototypical networks (Snell et al., 2017), the task-specific parameters are the mean vectors of same-class support samples, which can be computed as

$$\phi_{T_i} = \{\frac{1}{N} \sum_{(x_j, y_j) \in \mathcal{D}_{T_i}^{tr}, y_j = k} f_\theta(x_j)\}_{k=1}^K. \tag{6}$$

**Model-based meta-algorithms.** Some of model-based meta-algorithms avoid inner-task training by learning a meta amortization network $G$ parameterized by $\psi$ to generate task-specific parameters $\phi_{T_i}$ using the support set as inputs (Gordon et al., 2018b,a), i.e.,

$$\phi_{T_i} = G_\psi\big(f_\theta(\mathcal{D}_{T_i}^{tr})\big). \tag{7}$$

Both $\theta$ and $\psi$ are global parameters to be optimized in meta-training.

**Meta-algorithms with closed-form solvers.** Several meta-algorithms adopt a simple algorithm with convex objective function as inner-task algorithm such that the task-specific parameters $\phi_{T_i}$ have a closed-form solution (Bertinetto et al., 2019; Lee et al., 2019). For example, (Bertinetto et al., 2019) uses ridge regression as inner-task algorithm, and the closed-form solution is

$$\phi_{T_i} = (X_\theta^T X_\theta + \lambda I)^{-1} X_\theta^T Y. \tag{8}$$

For brevity, $X_\theta = \{f_\theta(x_j)\}_{j=1}^m$ and $Y = \{y_j\}_{j=1}^m$, where $(x_j, y_j) \in \mathcal{D}_{T_i}^{tr}$ (Bertinetto et al., 2019).

## 4. Methodology

From the unified perspective, the key constraint in designing a meta-algorithm is to find an inner-task algorithm which has an explicit analytical solutions which significantly limits the expressiveness of the inner-task algorithms. The key constraint is caused by the normal meta-training strategy that the optimization of meta-parameters $\theta$ in meta-update need differentiate through the inner-task adaptation. To relax this constraint, we propose an adaptation-agnostic meta-training strategy which makes no assumption on such dependency.

### 4.1 Adaptation-Agnostic Meta-Training

In particular, we do not enforce the optimization of meta-parameters to back-propagate the inner-task adaptation but propose to conduct these two steps separately and iteratively. In inner-task adaptation, the meta-parameters $\theta$ is fixed, and the support set is fed to the shared embedding network and used to train the task-specific predictor $\mathcal{A}_{\phi_{T_i}}$. In meta-update, the task-specific parameters $\phi_{T_i}$ are fixed and the query set is used to optimize the meta-parameters $\theta$. The iteration scheme is formulated as follows:

$$\text{Inner-task adaptation: Fix } \theta, \phi_{T_i} = \arg\min_{x_{\phi_{T_i}}} \mathcal{L}(\mathcal{D}^{tr}_{T_i}; \theta, x_{\phi_{T_i}}), \tag{9}$$

$$\text{Meta-update: Fix } \phi_{T_i}, \theta = \theta - l_\theta \nabla_\theta \mathcal{L}(\mathcal{D}^{ts}_{T_i}; \theta, \phi_{T_i}), \tag{10}$$

where $\theta$ refers to the meta-parameters, i.e., the global parameters shared by all the tasks, and $\phi_{T_i}$ refers to the task-specific parameters, i.e., the local parameters which are different among the tasks. We call this training strategy *adaptation-agnostic*, since in Eq. (9), it allows to use any inner-task algorithm with any optimization algorithm as long as the meta loss function $\mathcal{L}(\mathcal{D}^{ts}_{T_i}; \theta, \phi_{T_i})$ is differentiable w.r.t. $\theta$ given $\phi_{T_i}$, regardless of whether $\phi_{T_i}$ has an analytical expression w.r.t. $\theta$.

### 4.2 Inner-Task Algorithm

Without the requirement of an analytical solution, the choice of the inner-task algorithm is of great flexibility. Naturally, we come up with a neural network with a non-convex loss function, i.e., cross-entropy loss. Since there is no restriction on the optimization algorithm or the network architecture, we simply use a multilayer perceptron (MLP) trained by SGD as an inner-task algorithm. The inner-task adaptation can be formulated as:

$$\phi_{T_i} = \phi_{T_i} - l_{\phi_{T_i}} \nabla_{\phi_{T_i}} \mathcal{L}(f_\theta(\mathcal{D}^{tr}_{T_i}); \phi_{T_i}), \tag{11}$$

where $\phi_{T_i}$ is randomly initialized for each task. This may make the inner-task algorithm more flexible and less prone to overfitting, as verified in our experiments.

**An Ensemble Inner-Task Algorithm.** The generality and flexibility of the proposed adaptation-agnostic meta-training strategy enables us to apply a powerful algorithm as the inner-task algorithm. We come up with an ensemble which can combine the advantages of different types of inner-task algorithm. As an instantiation, we combine the mean-centroid classification algorithm of (Snell et al., 2017), initialization-based inner-task algorithm in ANIL [1] (Raghu et al., 2020) and MLP proposed by us (Eq. (11)) as the inner-task algorithm. As illustrated in Fig. 2, during inner-task adaptation, we train a bag of diverse algorithms separately with the embedded support set over the embedding space and obtain three independent predictors. Meta-update is performed by aggregating the predictions of all the predictors on the query set to obtain final predictions and then using the final predictions to update the shared meta-parameters $\theta$. See details in Appendix A.

---

1. Raghu et al. (2020) shows that the simplified MAML, i.e., ANIL, achieves the same performance as MAML (Finn et al., 2017). Hence, it suffices to only compare with MAML.

Table 1: Results of 5-way classification tasks using Conv-4 (the above set) and ResNet-18 (the below set) respectively. See the complete table in Appendix B.

| *mini*ImageNet test accuracy | | |
|---|---|---|
| Model | 5-way 1-shot | 5-way 5-shot |
| Matching Net (Vinyals et al., 2016) | $43.56 \pm 0.84$ | $55.31 \pm 0.73$ |
| Relation Net (Sung et al., 2018) | $49.31 \pm 0.85$ | $66.60 \pm 0.69$ |
| MAML (Finn et al., 2017) | $46.70 \pm 1.84$ | $63.11 \pm 0.92$ |
| Protonet (Snell et al., 2017) | $44.42 \pm 0.84$ | $64.24 \pm 0.72$ |
| MetaOptNet (Lee et al., 2019) | $49.20 \pm 0.42$ | $65.54 \pm 0.38$ |
| A2M (Mean-centroid + MLP+ Init-based) | $\mathbf{50.31 \pm 0.87}$ | $\mathbf{68.55 \pm 0.67}$ |
| Matching Net (Vinyals et al., 2016) | $52.91 \pm 0.88$ | $68.88 \pm 0.69$ |
| Relation Net (Sung et al., 2018) | $52.48 \pm 0.86$ | $69.83 \pm 0.68$ |
| MAML (Finn et al., 2017) | $49.61 \pm 0.92$ | $65.72 \pm 0.77$ |
| Protonet (Snell et al., 2017) | $54.16 \pm 0.82$ | $73.68 \pm 0.65$ |
| MetaOptNet (Lee et al., 2019) | $50.83 \pm 0.45$ | $71.01 \pm 0.38$ |
| A2M (Mean-centroid + MLP+ Init-based) | $\mathbf{57.04 \pm 0.84}$ | $\mathbf{75.65 \pm 0.71}$ |

## 5. Experiments

The experiments are designed to evaluate the instantiation of A2M introduced in Sec. 4.2 on standard and cross-domain few-shot classification tasks. We evaluate our method on *mini*ImageNet (Vinyals et al., 2016) with Conv-4 and ResNet-18 under the standard 5-way 1-shot and 5-way 5-shot settings. See implementation details and ablation study in Appendix B

### 5.1 Results for Few-shot Classification

*(1) Standard few-shot classification.* Referring to Table 1, for both 1-shot and 5-shot tasks, A2M achieves comparable or superior performance compared with state-of-the-art meta-algorithms. Remarkably using ResNet-18 in Table 1, A2M outperforms the best meta-algorithms by achieving 3% and 2% absolute increases in the 1-shot and 5-shot tasks respectively, demonstrating the effectiveness of A2M. *(2) Cross-domain few-shot classification.* As shown in Table 2, A2M achieves 2.5%, 13% absolute increase over MAML and PN respectively. The results verifies the generalization ability of A2M which can even generalize to new tasks with a domain shift.

Table 2: Results for a 5-way cross-domain classification task.

| *mini*ImageNet→CUB | | |
|---|---|---|
| | 5-way 1-shot | 5-way 5-shot |
| Matching networks (Vinyals et al., 2016) | $41.10 \pm 0.74$ | $53.07 \pm 0.74$ |
| Prototypical networks (Snell et al., 2017) | $42.71 \pm 0.78$ | $62.02 \pm 0.70$ |
| Relation net (Sung et al., 2018) | $40.74 \pm 0.76$ | $57.71 \pm 0.73$ |
| MAML (Finn et al., 2017) | $32.77 \pm 0.64$ | $51.34 \pm 0.72$ |
| D-MLP | $35.88 \pm 0.66$ | $57.78 \pm 0.76$ |
| A2M (Mean-centroid + MLP + Init-based ) | $\mathbf{43.55 \pm 0.80}$ | $\mathbf{64.63 \pm 0.82}$ |

## 6. Conclusion

In this paper, we provided a unified view on the commonly used meta-training strategy and proposed an adaptation-agnostic meta-training strategy that is more general, flexible and less prone to overfitting. In future work, we target to analyze the theoretical properties of the adaptation-agnostic meta-training strategy and explore more powerful inner-task algorithms.

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

# Appendix A. Methodology

## A.1 Applying an Ensemble Inner-Task Algorithm

As illustrated in Fig. 2, during inner-task adaptation, we train a bag of diverse algorithms $\{\mathcal{A}^e\}_{e=1}^E$ separately with the embedded support set and obtain $E$ predictors, i.e., $\{\mathcal{A}^e(\mathcal{D}_{T_i}^{tr};\theta)\}_{e=1}^E$. Next, meta-update is performed by aggregating the predictions of all the predictors on the query set to obtain final predictions and then using the final predictions to update the shared meta-parameters $\theta$. Formally, the meta-training procedure of A2M is formulated as follows,

$$\text{Inner-task adaptation: Fix } \theta, \text{for } e \in \{1, 2, \ldots, E\}, \phi_{T_i}^e = \arg\min_{x_{\phi_{T_i}^e}} \mathcal{L}(\mathcal{D}_{T_i}^{tr}; \theta, x_{\phi_{T_i}^e}),$$

$$\text{Meta-update: Fix } \{\phi_{T_i}^e\}_{e=1}^E, \theta = \theta - l_\theta \nabla_\theta \mathcal{L}(\mathcal{D}_{T_i}^{ts}; \theta, \{\phi_{T_i}^e\}_{e=1}^E). \tag{12}$$

**An instantiation of A2M.** A2M (Eq. 12) is a very general framework, and it can basically integrate any inner-task algorithm as inner-task algorithm for diverse purposes. Since this paper focuses on few-shot classification, we instantiate A2M with an ensemble of the mean-centroid classification algorithm of ProtoNets (Snell et al., 2017), the initialization-based inner-task algorithm as in MAML (Finn et al., 2017), and a two-layer MLP as inner-task classifier proposed by us (Eq. (11)) as the ensemble inner-task algorithm for meta-learning. Note that for the inner-task algorithm of MAML, we use the version as in ANIL (Raghu et al., 2020) which naturally combined in A2M framework. In addition, we choose to combine the mean-centroid classification algorithm and the initialization-based algorithm due to their complementary capabilities. The former has low model capacity but stable, while the latter has high model expressiveness but can easily overfit. The effectiveness of the proposed ensemble inner-task algorithm is empirically verified by our experiments in Sec. 5.

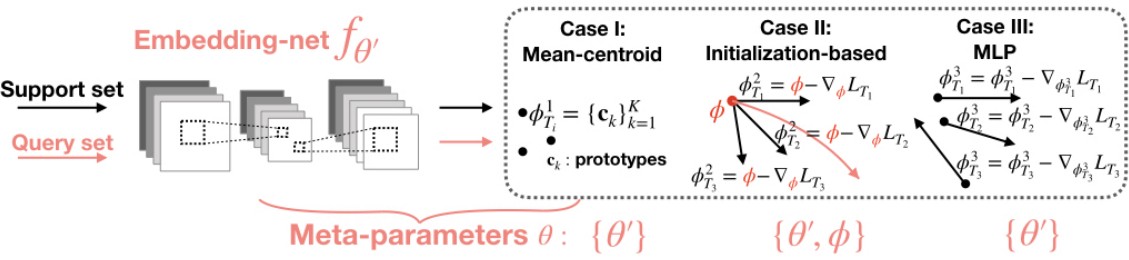

Figure 2: Inner-task adaptation of an instantiation (mean-centroid classification algorithm of (Snell et al., 2017), initialization-based inner-task algorithm in (Raghu et al., 2020) and MLP proposed by us (Eq. (11)) of A2M.

For inner-task adaptation, as illustrated in Fig. 2, the three algorithms are trained over the embedded support set independently, i.e,:

$$\mathcal{A}^1: \phi_{T_i}^1 = \{\mathbf{c}_k\}_{k=1}^K = \{\frac{1}{N}\sum_{z_j \in \mathcal{D}_{T_i}^{tr}, y_j=k} f_{\theta'}(x_j)\}_{k=1}^K,$$

$$\mathcal{A}^2: \phi_{T_i}^2 = \phi - l_\phi \nabla_\phi [\frac{1}{m}\sum_{z_j \in \mathcal{D}_{T_i}^{tr}} -\log\left(\frac{e^{g_\phi(f_{\theta'}(x_j))[y_j]}}{\sum_{k'} e^{g_\phi(f_{\theta'}(x_j))[k']}}\right)],$$

$$\mathcal{A}^3: \phi_{T_i}^3 = \phi_{T_i}^3 - l_{\phi_{T_i}^3} \nabla_{\phi_{T_i}^3} [\frac{1}{m}\sum_{z_j \in \mathcal{D}_{T_i}^{tr}} -\log\left(\frac{e^{g_{\phi_{T_i}^3}(f_{\theta'}(x_j))[y_j]}}{\sum_{k'} e^{g_{\phi_{T_i}^3}(f_{\theta'}(x_j))[k']}}\right)], \tag{13}$$

where $\mathcal{A}^1$, $\mathcal{A}^2$ and $\mathcal{A}^3$ denote the mean-centroid classification algorithm, the initialization-based algorithm and the two-layer MLP, respectively. Note that for $\mathcal{A}^2$, $\phi$ is shared by each task and updated during meta-update.

In meta-update, as shown Fig. 2, for any query $\{z_j = (x_j, y_j) \in \mathcal{D}_{T_i}^{ts}\}$, the predictions of $\mathcal{A}^1$, $\mathcal{A}^2$ and $\mathcal{A}^3$ are aggregated to produce the final prediction. In our instantiation, we sum up all the predictions and use the output as the query's logits for computing the cross-entropy loss. Specifically, given the task-specific parameters $\phi_{T_i}^1$, $\phi_{T_i}^2$ and $\phi_{T_i}^3$, the meta-update process is as follows,

$$\theta = \theta - l_\theta \nabla_\theta [\frac{1}{q}\sum_{z_j \in \mathcal{D}_{T_i}^{ts}} -\log\left(\frac{e^{g_{\phi_{T_i}^3}(f_{\theta'}(x_j))[y_j]+g_{\phi_{T_i}^2}(f_{\theta'}(x_j))[y_j]-d(f_{\theta'}(x_j),\mathbf{c}_{y_j})}}{\sum_{k'} e^{g_{\phi_{T_i}^3}(f_{\theta'}(x_j))[y_j]+g_{\phi_{T_i}^2}(f_{\theta'}(x_j))[k']-d(f_{\theta'}(x_j),\mathbf{c}_{k'})}}\right)],$$

where $d(\cdot, \cdot)$ is the distance between the query's embedding and the prototype and $\theta = \{\theta', \phi\}$.

## Appendix B. Experiments

### B.1 Experimental Setup

#### B.1.1 DATASETS

The ***mini*ImageNet** (Vinyals et al., 2016) consists of 100 classes with 600 images per class. The dataset is split into a training set with 64 classes, a testing set with 20 classes and a validation set with 16 classes (Ravi and Larochelle). Following the convention, the images are cropped into $3\times84\times84$ and $3\times224\times224$ when using CNN-based (Vinyals et al., 2016) and ResNet-based model architectures (Chen et al., 2019) respectively.

The **CUB** dataset (Wah et al., 2011) contains 200 classes and 11,788 images in total. The CUB dataset is split into 100 classes for training, 50 classes for validation and 50 classes for testing (Chen et al., 2019). The input size of images in CUB is $3\times224\times224$.

#### B.1.2 IMPLEMENTATION DETAILS

In order to achieve a **fair** comparison, we employ the consistent experimental environment proposed in (Chen et al., 2019) and strictly follow the training details in it. Specifically,

we compare the performance using the widely-used Conv-4 as in (Snell et al., 2017) and the ResNet-18 backbone adopted in their environment. We have not applied any high-way or high-shot training strategy. For the optimizer, we use Adam (Kingma and Ba, 2014) as the meta-optimizer with a fixed learning rate 0.001. For the cross-domain tasks, we train models on the entire *mini*ImageNet dataset. The meta-validation and meta-test of the models use the validation set and test set of the CUB dataset respectively.

### B.1.3 Comparison with the state-of-the-art

For **fair** comparison, here we compare with the state-of-the-art methods that have a similar implementation (e.g., using the same backbone network) as ours. In this paper, we use a standard ResNet-18 backbone (He et al., 2016). Differently, MetaOptNet (Lee et al., 2019) and TADAM (Oreshkin et al., 2018) use a ResNet-12 backbone; LEO (Rusu et al., 2018) uses a WRN-28-10 backbone. Besides, we do not use techniques such as DropBlock regularization, label smoothing and weight decay as adopted in MetaOptNet (Lee et al., 2019) to increase performance. Hence, we do not compare with these methods.

## B.2 Main Results

### B.2.1 Performance on *mini*Imagenet.

For the standard few-shot scenario, we conduct experiments of 5-way 1-shot and 5-way 5-shot classification on *mini*ImageNet with the Conv-4 and the ResNet-18 backbones. The results are shown in Table 1. For both 1-shot and 5-shot tasks, our model achieves comparable or superior performance compared with state-of-the-art meta-algorithms. Remarkably on the ResNet-18 backbone in Table 1, A2M outperforms the best meta-algorithms by achieving approximate 3% and 2% absolute increases in the 1-shot and 5-shot tasks respectively, demonstrating the effectiveness of A2M.

### B.2.2 Cross-domain classification.

To further examine the generalization ability of our method, we conduct experiments on the challenging cross-domain classification task proposed in (Chen et al., 2019). The results are shown in Table 2. Here, D-MLP denotes the decoupled meta-training with a MLP inner-task algorithm proposed by us as in Sec. 4.2. For 5-way 5-shot classification, D-MLP achieves 6% absolute increases compared with MAML (Finn et al., 2017), which indicates that D-MLP is less prone to overfitting than MAML. Our ensemble framework A2M achieves 7%, 2.5%, 13% absolute increase over D-MLP, MAML (Finn et al., 2017) and PN (Snell et al., 2017) respectively. The results show that our adaptation-agnostic ensemble framework facilitates the meta-net to learn more general structures that can adapt better to new tasks with a domain shift.

## B.3 An Ablation Study of A2M

To further study our proposed A2M, we provide an ablation study using the Conv-4 backbone. In Table 4, we can observe that A2M (mean-centroid + MLP + init-based) achieves best results when compared with each individual component or an ensemble of any two components. This further demonstrates that the ensemble method is effective.

Table 3: Results of 5-way classification tasks using Conv-4 (the above set) and ResNet-18 (the below set) respectively. Compared results are from references except ∗ re-implemented by (Chen et al., 2019).

| *mini*ImageNet test accuracy | | |
|---|---|---|
| **Model** | 5-way 1-shot | 5-way 5-shot |
| Matching Net (Vinyals et al., 2016) | $43.56 \pm 0.84$ | $55.31 \pm 0.73$ |
| Relation Net (Sung et al., 2018) ∗ | $49.31 \pm 0.85$ | $66.60 \pm 0.69$ |
| Meta LSTM (Ravi and Larochelle) | $43.44 \pm 0.77$ | $60.60 \pm 0.71$ |
| SNAIL (Mishra et al., 2018) | $45.10$ | $55.20$ |
| LLAMA (Grant et al., 2018) | $49.40 \pm 1.83$ | $-$ |
| REPTILE (Nichol and Schulman, 2018) | $49.97 \pm 0.32$ | $65.99 \pm 0.58$ |
| PLATIPUS (Finn et al., 2018) | $50.13 \pm 1.86$ | $-$ |
| GNN (Garcia and Bruna, 2018) | $50.30$ | $66.40$ |
| R2-D2 (high) (Bertinetto et al., 2019) | $49.50 \pm 0.20$ | $65.40 \pm 0.20$ |
| MAML (Finn et al., 2017) ∗ | $46.70 \pm 1.84$ | $63.11 \pm 0.92$ |
| Protonet (Snell et al., 2017) ∗ | $44.42 \pm 0.84$ | $64.24 \pm 0.72$ |
| MetaOptNet (Lee et al., 2019) ∗ | $49.20 \pm 0.42$ | $65.54 \pm 0.38$ |
| A2M (Mean-centroid + MLP+ Init-based) | $\mathbf{50.31 \pm 0.87}$ | $\mathbf{68.55 \pm 0.67}$ |
| Matching Net (Vinyals et al., 2016) ∗ | $52.91 \pm 0.88$ | $68.88 \pm 0.69$ |
| Relation Net (Sung et al., 2018) ∗ | $52.48 \pm 0.86$ | $69.83 \pm 0.68$ |
| MAML (Finn et al., 2017) ∗ | $49.61 \pm 0.92$ | $65.72 \pm 0.77$ |
| Protonet (Snell et al., 2017) ∗ | $54.16 \pm 0.82$ | $73.68 \pm 0.65$ |
| MetaOptNet (Lee et al., 2019) ∗ | $50.83 \pm 0.45$ | $71.01 \pm 0.38$ |
| A2M (Mean-centroid + MLP+ Init-based) | $\mathbf{57.04 \pm 0.84}$ | $\mathbf{75.65 \pm 0.71}$ |

Table 4: An ablation study about components in A2M using the Conv-4 backbone. Results are obtained on *mini*Imagenet.

| Mean-centroid | MLP | Init-based | 5-way 1-shot | 5-way 5-shot |
|:---:|:---:|:---:|---|---|
| ✓ | | | $44.42 \pm 0.84$ | $64.24 \pm 0.72$ |
| | ✓ | | $45.00 \pm 0.39$ | $64.38 \pm 0.33$ |
| | | ✓ | $46.70 \pm 1.84$ | $63.11 \pm 0.92$ |
| ✓ | ✓ | | $46.99 \pm 0.43$ | $66.61 \pm 0.38$ |
| | ✓ | ✓ | $49.74 \pm 0.88$ | $63.84 \pm 0.73$ |
| ✓ | | ✓ | $50.10 \pm 0.81$ | $67.55 \pm 0.37$ |
| ✓ | ✓ | ✓ | $\mathbf{50.31 \pm 0.87}$ | $\mathbf{68.55 \pm 0.67}$ |

Besides, we observe that A2M has the advantage of combining the strength of individual components while mitigating their drawbacks. On one hand, focusing on the results of 5-way 1-shot classification. It can be seen that all the variants of A2M achieve better results compared with the individual component. It is well known that models are extremely easy

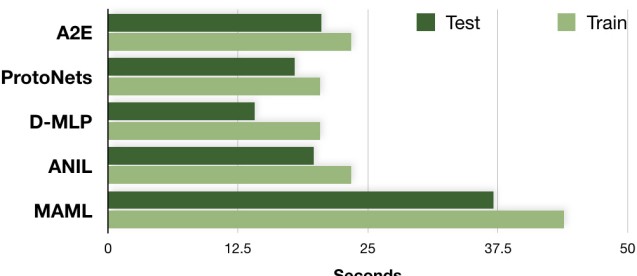

Figure 3: Comparison of running times. For brevity, A2M is short for A2M (mean-centroid + D-MLP + init-based) and MAML is the abbreviation for MAML first-order approximate version

to overfit in the 1-shot scenario. Clearly, our framework is capable of reducing classification variance in such cases. On the other hand, inspecting the outcomes of the 5-way 5-shot classification, the results of the ensemble including the mean-centroid component are 66.61% and 67.55% which outperform the result of (MLP+init-based), i.e., 63.84% without the mean-centroid component. It demonstrates the power of the mean-centroid component in preventing overfit as the shot number increases and the ensemble method can obtain such advantage after incorporating the mean-centroid classification algorithm.

### B.4 Efficiency

We provide a quantitative comparison by measuring the meta-training and meta-testing time for 100 episodes shown in Fig. 3 and Our results are obtained on 5-way 1-shot models with the ResNet-18 backbone. Fig. 3 shows that our A2M merely increases the running time by a small margin even when combining three components in the ensemble and validates our statement that A2M is efficient.

