# OpenReview forum: "Adaptation-Agnostic Meta-Training"
_ICML.cc/2021/Workshop/AutoML — AutoML@ICML2021 Poster_

### Official Review · Reviewer_aNA2 · 2021-06-06
**Acceptable paper on meta-learning**

**Rating:** 7
**Confidence:** 4

**Review:**

This paper presents an approach to use any base learning method, including ensembles of them, in the inner loop of meta-learning algorithms. The idea is to fix the parameters learned by a base learner when performing the meta-update. The paper is clearly in the scope of the workshop and may generate interesting discussions about algorithms and objectives in meta-learning.

For longer term impact there many areas that the paper could be improved if the authors seek longer term impact:
- The mathematical presentation is not very clear. I was not able to find the meta-loss being differentiated to obtain the meta-update.
- The general idea seems very similar to algorithms such as first-order MAML and Reptile, which while using SGD as the base learner do not depend on the ability to take a derivative through the algorithm to derive the meta-update. The relationship between the methods should be discussed.
- Experimentally, there are missing comparisons of the ensembling techniques to natural baselines such as using simple ensembling techniques on top of separately-trained regular meta-learning algorithms.

---

### Official Review · Reviewer_5tLt · 2021-06-13
**Interesting meta-learning approach, with encouraging results. Writing, comparisons, and claims can be improved.**

**Rating:** 6
**Confidence:** 4

**Review:**

## quality

### + pros

- Clear positioning of idea among existing meta-learning algorithms

- In Table 4, it is interesting that a randomly initialized (!) 2 layer MLP head with only 1 gradient step (Equation 13) can have/add significantly to model performance, and at least not hurt it. Assuming the correctness of the experiments, this is a very interesting result.

### - cons

> "task-params should be an analytical expression of meta-params"
>
> "only simple algorithms with closed-form solvers can be applied as the inner-task algorithm, such as ... SVM (Lee et al., 2019)"

- SVM (Lee et al., 2019), while simple, is not closed-form; uses 3 iterations

Also, there are analytical examples where the inner-task algorithm is not simple and/or closed-form. E.g.:

- LEO (Rusu et al., 2018), where $\mathcal{A}$ is a generative model, is not a simple inner-task algorithm



> "less prone to overfitting"

- conclusion without data; comparison train/test accuracy not present. Only test data is not enough.

\_

- The paper claims a *fair* comparison, by following (Chen et al., 2019), but omits its important (non meta-learning) baseline/baseline++ in the results.

## clarity

### + pros

- good organization of sections

### - cons

- a lot of undefined terminology (see below)
- writing and English should be improved



Equation (1):

- $\mathcal{L}$ is undefined
- no indexes on $\theta$, makes the equation mathematically incorrect
- learning rate = 1, or no learning rate? See, e.g., MAML (Finn et al., 2017)

Equation (3): $l$ undefined

Equation (5): $l_{\phi_{T_i}}$ undefined

Equation (6): $N$ undefined

"As illustrated in Figure 2": Figure 2 is missing in main manuscript.

Table 1: do better separation/mentioning (inside table) between Conv-4 and ResNet-18

Table 2: "D-MLP" undefined

"PN" undefined

## originality

### + pros

- Novel and general way of not needing to differentiate through the task adaptation process, by only relying on task-specific parameters and their non-adaptation relation to the meta-parameters in the predictions.
- Interesting idea to ensemble multiple inner-task algorithms to get best of many worlds

## significance

### + pros

- Consistent improvements over popular old meta-learning baselines
- Allows for more powerful inner-task algorithms

### - cons

- Marginal improvements over popular old meta-learning baselines
- No experimental comparison with non meta-learning baselines (e.g., Chen et al., 2019), nor with more recent meta-learning work such as Meta-Baseline [1].

## typos

algotithm, A2E (Figure 3)



## references

[1] A New Meta-Baseline for Few-Shot Learning, 2020

---

### Official Review · Reviewer_DWny · 2021-06-17
**Review for Adaptation-Agnostic Meta-Training**

**Rating:** 4
**Confidence:** 3

**Review:**

This paper aims to propose a general meta-learning framework with multiple algorithms. The author unified recent meta-learning researches and proposed a decompled training meta-learning approach to combine different algorithms. First, they proposed to update the task/algorithm-specific parameters in the inner loop. Then, they update the shared feature extraction network in the outer loop. Compared to the previous work, this approach can combine with different metric/gradient-based meta algorithms. Finally, they compare the related meta-learning algorithms on MiniImagenet and show the cross-domain performance from MiniImagenet to CUB. According to their result, their approach has a comparable result on computer vision datasets.

However, the novelty of this paper is limited. The unified view has already been summarized in a previous paper [1]. Besides, the proposed approach seems to be an incremental improvement of ANIL.

Pros:
- The author shows good performance on some computer vision dataset.

Cons:
- The proposed unified view already summarised in previous published paper [1].
- Apart from the proposed unified view, the main contribution of this paper seems to be an incremental improvement from the previous ANIL method.
- Some related competitor results are missing. It will be better to include the algorithm VERSA.

Minor:
- The English presentation needs to improve.
- In equation 3, x_j, y_j should be contained in z_j. But this is not easily readable from the equation.

[1] Jonathan Gordon, John Bronskill, Matthias Bauer, Sebastian Nowozin, and Richard E Turner. Meta-Learning Probabilistic. Inference for Prediction. ICLR 2019

---

### Decision · Program_Chairs · 2021-06-21

Accept (Poster)